# A scoping review key elements and effects of cardiovascular disease management programs based on community-based participatory research

**Juhyeon Yang**, **Bohyun Park** *

Department of Nursing, Changwon National University, Changwon, Gyeongnam, Republic of Korea

* bhpark@changwon.ac.kr

## Abstract

### Background

This scoping review analyses the literature on community-based participatory research (CBPR)–based cardiovascular disease (CVD) management programmes, examining the key elements of their development and implementation and exploring their effectiveness.

### Methods

This scoping review's methodology had six stages: 1) identifying the research question; 2) identifying relevant studies—search strategy; 3) study selection; 4) charting the data; 5) collating, summarising, and reporting the results; and 6) consultation exercise. The databases used were PubMed, Cochrane, and CINAHL, for the period from 4 March to 3 April 2022. We selected studies 1) published after 2000; 2) targeting community residents over 18 years old; and 3) proposed a CBPR-based CVD management programme, described its development, and evaluated its effects based on its application. Data were extracted independently by each of the two researchers, using a standardised form.

### Results

Among the key aspects of such programmes were the many cases where community organisations led establishment of partnerships and cases where a decision-making committee was formed. Regarding application of the CBPR principles, community partners participated only in executing the research, not in analysing and interpreting research results. In addition, among the 21 studies selected were 6 randomised controlled trials, all of which showed a significant positive effect in experimental groups compared to control groups.

### Conclusion

Improvement strategies are needed to allow implementation of CBPR principles in a CBPR-based CVD management programme. Moreover, further verification of programme evaluation research methods is needed.

**Data Availability Statement:** All relevant data are within the paper and its Supporting information files. The data generated in this study was

disclosed by uploading to figshare dataset. https://doi.org/10.6084/m9.figshare.20280627.v1.

**Funding:** This study was financially supported by National Research Foundation of Korea (No: 2020R1A2C1008591). The funding agency had no influence on the design, data collection, analysis, interpretation of data, or results of this study, nor on the writing of this manuscript.

**Competing interests:** The authors declare no conflict of interests.

**Abbreviations:** BP, Blood pressure; CBO, Community-based organization; CBPR, Community-based participatory research; CHW, Community health worker; Con., Control group; CVD, Cardiovascular disease; DM, Diabetes mellitus; Exp., Experimental group; FBO, Faith-based organization; HDL, High density lipoprotein; HE, Health educator; HPLP, Health promotion lifestyle profile; LDL, Low density lipoprotein; LS7, Life's Simple 7; NR, Not reported; PACES, Physical Activity Enjoyment Scale; Qual, Results of Qualitative research; Quan, Results of Quantitative research; RCT, Randomised controlled trial; RR, Relative risk.

## Scoping review registration

This protocol has been registered to the OSF registries. 0000000204460911. Key Elements and Effects of Cardiovascular Disease Management Programs Based on Community-based Participatory Research: Protocol for a Scoping Review'. OSF, 4 Sept. 2020. Web.

## Introduction

According to the World Health Organization, cardiovascular disease (CVD) is one of the most common causes of death, with 18 million annual deaths globally [1], and poses an enormous burden to global health and the global economy [2]. In many countries, the mortality rate of CVD is higher among people from lower socioeconomic strata [3, 4]. Such differences in the mortality rate of CVD according to income level arise because the factors affecting the occurrence of CVD and/or the resultant mortality differ depending on income level [3, 5, 6]. Specifically, access to resources promotes healthy behaviour and is regarded as a key social determinant of health [7]. Children from families of low socioeconomic status are more likely to live in areas where such access and opportunities are scarce [8]. In other words, unequal distribution of social determinants of health leads to inequalities in the overall health levels. The need to reduce health inequalities is an indisputable agenda, for which various efforts are being made worldwide. Community-based participatory research (CBPR) is proposed as one such effort.

CBPR engages community members, organisational representatives, and academic researchers equally in research [9]. The core of CBPR lies in establishing equal, cooperative health promotion partnerships among community participants based on the principle of equality of participation among the different members of society [10]. As these equal participants independently identify problems and come up with ideas for solutions in the community, it grants the method validity and sustainability, demonstrating its utility in practice [11]. Such a process can help establish and reinforce the network of related organisations in the local community and improve leadership, which in turn can enhance the autonomous, sustainable capacity for health promotion in the local community [12]. While 'participation' from members of the local community is emphasised during the CBPR process, similar weight is placed on the concept of 'practice'. In other words, CBPR integrates achievement using community capacity and networks with practice-oriented activities [13]. Therefore, CBPR can effectively act as a liaison in the dissemination of academic research results to health project sites, for instance.

The community capacity that CBPR aims to foster is regarded as a factor mediating the gap between socioeconomic factors and disparities in health levels, which is among the mechanisms of health inequalities. A CBPR-based health promotion programme is based on these mechanisms of health determinants and aims to reinforce community capacity using mediating variables such as social networks, social support, and social capital, thereby reducing and ultimately eliminating health inequalities and improving health status. In particular, in order to reduce risk factors for chronic disease such as CVD and eliminate health inequality caused by it, more emphasis is being placed on the importance of the social determinants of health, which is defined as the social, environmental, and related factors in which an individual is born, grows, and resides [14]. The American Heart Association pointed out that 'the most significant opportunities for reducing deaths and disability from CVD in the United States lie with addressing the social determinants of cardiovascular outcomes' [15]. Therefore, the

CBPR-based CVD management programme is expected to improve CVD health status compared to existing programmes. However, as community capacity cannot be reinforced in a short span of time, it may take considerable time to achieve improved health status. Nevertheless, reinforced community capacity cannot be easily lost and is, therefore, expected to have long-lasting effects, unless the members of the community change.

Reviews of studies implementing CBPR in the field of health care revealed that they targeted specific population groups (Hispanic, African-American, Asian-American, immigrants) [16, 17] or residents living in specific regions such as Sub-Saharan Africa and islands in the Asia Pacific region [18, 19] and focused on the mechanisms of resident participation. Meanwhile, the literature is lacking in reviews that focus on synthesising and presenting the effects of CBPR-based health programmes.

Our scoping review aims to analyse the literature on CBPR-based CVD management programmes, examining the core elements of their development and implementation and exploring their effectiveness.

## Materials and methods

This study is based on the scoping review methodology developed by Arcsey and O'Malley [20] and modified by Levac et al. [21]. It consists of six stages: 1) identifying the research question; 2) identifying relevant studies—search strategy; 3) study selection; 4) charting the data; 5) collating, summarising, and reporting the results; and 6) consultation exercise. In the process of conducting this scoping review and reporting the results, we abided by the PRISMA-ScR (Preferred Reporting Items for Systematic Reviews and Meta-Analyses Extension for Scoping Reviews) checklist [22].

### Stage 1: Identifying the research questions

The research questions in this scoping review were formulated and divided into two categories: 1) the 'process' aspect of developing and implementing a CBPR-based CVD management programme, explored through a preliminary survey; and 2) the 'results' aspect of developing and implementing the programme. The research questions are as follows:

1. What are the key elements to be considered in the process of developing and implementing a CBPR-based CVD management programme?

2. How is the CBPR-based CVD management programme structured, and what are its effects?

### Stage 2: Identifying relevant studies—The search strategy

The electronic databases used for the literature search were PubMed, Cochrane, and CINAHL. The first search was conducted in April 2020. Later, the literature published from April 2020 to April 2022 was also searched.

The search terms were selected by reviewing the main terms used in the abstract or main text of relevant studies. Based on the search field characteristics for each electronic database, a specific search strategy was established. The search strategies used across all the electronic databases are as follows: 1) The 'population' search words include 'cardiovascular disease', 'vascul* disease', and 'vascular disease', combined using the Boolean operator OR; 2) The 'intervention' search words include 'community–based participatory research', 'participatory action research', 'CBPR', 'PAR', 'participant*', 'community engagement', 'community involvement', 'civic engagement', and 'engagement*', combined using the Boolean operator OR; 3)

The population and intervention search results were combined using the Boolean operator AND; 4) For a comprehensive search, a Medical Subject Headings (MeSH) search was first performed for all search words, followed by a search performed in the title/abstract field; 5) The search results were saved after removing duplicate literature using bibliographic management software (EndNote V.9.3.3); 6) The search results were shared among the authors.

## Stage 3: Study selection

In order to elicit answers for the research questions of this scoping review, we determined the selection criteria for this review, as shown in Table 1, and conducted study selection using those criteria. A study was selected if two researchers reached a consensus after independently screening the study title and abstract. When it was difficult to select a study based solely on the abstract, the full text was reviewed to make a final decision. When the two researchers disagreed, the study selection was made or not made after discussion between them.

A flow chart of the study selection process is shown in Fig 1. Among the 877 studies identified during the preliminary search of electronic databases, 72 duplicate studies were removed. Upon reviewing the titles and abstracts of the remaining 805 studies, 663 studies were excluded from the analysis because they did not meet the selection criteria. We selected 28 studies after reviewing the full text of 142 studies; the remaining 114 studies did not meet the selection criteria. The following groups of studies were derived from the same research and hence, are considered as one study: 1) Balcazar, Rosenthal et al. [23], Balcazar, de Heer et al. [24], and Balcazar, Wise et al. [25]; 2) Brewer, Balls-Berry et al. [26] and Brewer, Morrison et al. [27]; 3) Brewer, Hayes, Caron et al. [28] and Brewer, Hayes, Jenkins et al. [29]; and 4) Ralston, Lemacks et al. [30], Ralston, Young-Clark et al. [31], and Ralston, Wickrama et al. [32]. Ultimately, the number of studies included in the review was 22.

## Stage 4: Charting the data

After the data were extracted independently by two researchers according to the standardised form, the results of the extraction process were compared. A data extraction form was prepared before the start of data extraction and was adjusted as references were compiled on a pilot basis. Through regular research meetings for 2–3 hours twice a week, the results of data extraction were checked and the degree of agreement was compared.

**Table 1. Inclusion and exclusion criteria.**

|  | **Inclusion criteria** | **Exclusion criteria** |
|---|---|---|
| Publication date | Published in or after 2000 |  |
| Study design | Quantitative study (randomized trial, clustered randomized trial, non-randomized trial, repeated measures study, cohort study, case-control study, interrupted time series study, controlled before–after study, before–after study) Qualitative study Mixed method | Quantitative study (non-comparative study) Non–original study (reviews, letters, opinions etc.) |
| Participants | Community residents over 18 years old | Patients/clients visit or are admitted to facilities, clinics, or hospitals |
| Intervention | CVD management program based on CBPR Including at least one among the five components composing the CBPR quality assessment tool developed by Viswanathan and colleagues [20] and revised by Chen and colleagues [21] |  |

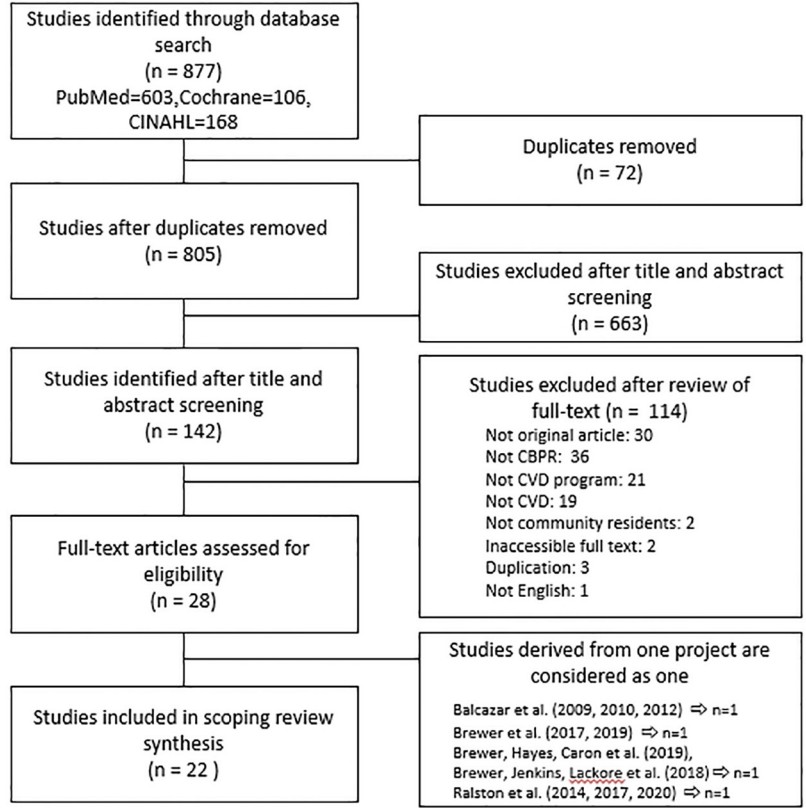

**Fig 1. The sequential process of identifying the studies included in the analysis.**

## Stage 5: Collating, summarising, and reporting the results

For the results regarding Research Question 1—the key factors to be considered in the process of developing and applying a CBPR-based CVD management programme—we summarised common characteristics observed in selected studies, such as partnership establishment (the subject of research initiation and the structure of partnership) and the process of development and application of a CVD management programme (the degree of CBPR principle application, application of planning model and theory, and field activists). We referred to the CBPR quality assessment tool developed by Viswanathan et al. [33] and revised by Chen et al. [34] for the degree of CBPR principle applications, subsequently employing the questions recommended by the latter. The tool modified by Chen et al. [34] consists of two questions as the primary criteria of assessment—'Was the community partner identified?' and 'Was the community partner involved in the planning and/or execution of research?'—and three questions for the secondary criteria—'Was the community partner involved in selection of research topic or development (or review) of the programme?', 'Was the community partner involved in analysis and/or interpretation of research?', and 'Was the community partner involved in dissemination of research results?'. Each question is measured on a three-point scale (1 = 'poor'; 2 = 'fair'; and 3 = 'good'). Since all studies reported that there were community partners, the question on whether community partners were identified was removed. Regarding the question on whether the community partner was involved in selection of research topic or development (or review) of the programme, we considered participants from the community who participated in the process of selecting a topic or changing an existing programme to suit their

characteristics to be community partners. The results were summarised by evaluating each question, not on the aforementioned three-point scale, but on a simplified two-point scale (yes and no).

For Research Question 2—'How is the CBPR-based CVD management programme structured and what is its effects?'—we summarised the study design, characteristics of participants and intervention programmes, measurement scales, and results. The effects of the programme were listed in the order of study design [single group pre–post, non-randomised controlled trial (non RCT), randomised controlled trial (RCT), and qualitative study] and were described qualitatively.

## Stage 6: Consultation exercise

In order to achieve the purpose of this scoping review, the authors received expert advice on the results derived through the final step of the scoping review, namely reviewing. Three experts with experience in CBPR were asked to review the results. The experts suggested their opinions on whether the situation or experience in CBPR sites was properly reflected in the review result, and the researchers reflected the opinions of the experts during the process of data extraction and summarising of results.

## Results

### General characteristics of selected studies

First, the selected 21 studies were divided into the following main areas of research: purpose, method, country in which the study was conducted, participants, and health problems of participants (Table 2). As for purpose of study, there were four studies (18.2%) on the

**Table 2. General characteristics of selected study.**

| | | (N = 22) |
| --- | --- | --- |
| | | n (%) |
| Purpose of study | Developing process of CVD management program | 4 (18.2) |
| | Development and effectiveness evaluation of CVD management program | 4 (18.2) |
| | Effectiveness evaluation of pre-developed CVD management program | 14 (63.6) |
| Research methodology | Quantitative study | 10 (45.5) |
| | Qualitative study | 6 (27.3) |
| | Mixed method | 6 (27.3) |
| Race | African American (AA) | 8 (36.4) |
| | Hispanic | 5 (22.7) |
| | African | 2 (9.1) |
| | Asian | 1 (4.5) |
| | Not reported | 6 (27.3) |
| Participants | Male and female adults | 17 (77.3) |
| | Only female adults | 2 (9.1) |
| | Nor reported | 3 (13.6) |
| Health problem of participants | Healthy people | 1 (4.5) |
| | Having one or more cardiovascular risk factors | 8 (36.4) |
| | Nor reported | 13 (59.1) |

development process of a CBPR-based CVD management programme, four studies (18.2%) on the development, investigation, and evaluation of a CBPR-based CVD management programme, and 14 studies (63.6%) that evaluated a CBPR-based CVD management programme. As for the research method, quantitative studies accounted for the largest proportion, with ten studies (45.5%), followed by six qualitative studies (27.3%) and six mixed-methods studies (27.3%). As for the race of the participants in the study, studies on African Americans accounted for the largest proportion (36.4%), followed by five studies on Hispanics (22.7%), two studies on Africans (9.1%), and one study on Asians (4.5%). In addition, six studies (27.3%) did not mention the race of participants. In one of the studies, firefighters were selected as the study subjects, and in the other studies, community residents were selected as the study subjects. There were 17 studies (77.3%) on adult males and two studies (9.1%) on only adult females. The remaining three studies (13.6%) did not mention the gender of the participants. In terms of the participant selection criteria, there was one study (4.5%) on healthy subjects without any underlying disease and eight studies (36.4%) on subjects with underlying diseases. The underlying diseases were further classified into high blood pressure, overweight or obesity, diabetes, and other CVD risk factors.

### Key elements of CBPR-based CVD management programme

**Establishing partnership.** *Subject of research initiation*. In four of the selected studies, local residents initially proposed the study voluntarily. In Bradley and Puoane [35], the research began once the local community health worker (CHW) received complaints from local residents and consulted local health experts. In a study by Delisle et al. [36], firefighters took the initiative to establish a cooperative relationship with the local community. In Lynch et al. [37], the pastor of an African American church requested help from local colleges to improve the health care of the church members, which subsequently led to the establishment of a cooperative relationship. In Ralston et al. [30–32], cooperative relationships with local colleges for health promotion programmes were established, led by a church in Northern Florida.

*Partnership establishment status*. With 13 studies, the most common forms of partnership were between community members, through community-based organisations (CBO) or faith-based organisations (FBO), and researchers from universities (Table 3). Without CBOs or FBOs, CHW, public institutions, and medical institutions played the leading role. In the nine studies, the resident representatives participated as a member of the partnership. However, there were very few studies (two) in which only resident representatives participated, without local community organisations. In the nine studies, after establishing community partnerships, a steering committee or advisory board was formed to carry out the project, which made decisions through regular meetings. In some studies, a working-level meeting was held by forming a subcommittee during the programme development process.

**Development and implementation of intervention programmes.** *Application of CBPR principles*. CBPR quality evaluation criteria (whether community partners participated in the process of planning and conducting research, selection and modification of research topics, and analysis/interpretation and dissemination of research results) were used to analyse how consistently the CBPR principles were applied during the process of development and implementation of intervention programmes (Table 4). While it has been observed that community partners participated in the process of conducting research in all studies, they participated in the process of research planning only in 16 studies. Among 16 studies in which community partners participated in the research topic selection or modification process, there was no study in which community partners participated in the analysis and interpretation of the research results. In 10 studies, community partners participated in the process of

**Table 3. Establishing partnership.**

| | Author (year) | Community partner | | | | | | | |
|---|---|---|---|---|---|---|---|---|---|
| | | University | CBO | FBO | Public organisation | Others | Community members | Committee (or advisory board) | Community activist |
| CBO/FBO + University | Balcazar, Rosenthal et al. 2009 [23]<br>Balcázar, de Heer et al. 2010 [24]<br>Balcázar, Wise et al. 2012 [25] | O | O | | | Hospital | | O | Promotores |
| | Brewer, Hayes, Caron et al. 2019 [28]<br>Brewer, Jenkins, Lackore et al. 2018 [29] | O | | O | | | | | FAITH partner |
| | Ralston, Lemacks et al. 2014 [30]<br>Ralston, Young-Clark et al. 2017 [31]<br>Ralston, Wickrama et al. 2020 [32] | O | | O | | | O | | |
| | Bess et al. 2019 [38] | O | O | O | | | O | O | |
| | Brewer, Balls-Berry et al. 2017 [26]<br>Brewer, Morrison et al. 2019 [27] | O | | O | | | | | FAITH partner |
| | Jayaprakash et al. 2016 [39] | O | O | | | | | O | |
| | Kim et al. (2004) [40] | O | O | | | | | O | Lay health advisors |
| | Lynch et al. (2019) [37] | O | | O | | | O | O | |
| | Mudd-Martin et al. (2013) [41] | O | O | | | | | | Promotores |
| | Pazoki et al. (2007) [42] | O | O | | | | | O | Local health volunteers |
| | Pinsker et al. (2017) [43] | O | | O | | | O | | |
| | Schulz et al. (2015) [44] | O | O | O | | | O | | Community health promotors |
| | Skolarus et al. (2018) [45] | O | | O | | | | | |
| Others + University | Bradley & Puoane (2007) [35] | O | | | | CHW | | | |
| | Chimberengwa & Naidoo (2019) [46] | | | | O | Nurse | O | O | Village health workers |
| | Delisle et al. (2013) [36] | O | | | O | | O | | Peer fitness trainer |
| | Rorie et al. (2011) [47] | O | | | O | | | | Resident health advocates |
| | Schroeder et al. (2017) [48] | O | | | | | O | | |
| | Folta et al. (2019) [49] | O | | | | HE | | | Local leader |
| | Guzman-Tordecilla (2002) [50] | O | | | | Nurse, expert | O | | Promoters |
| Community members + University | Perry et al. (2017) [51] | O | | | | | O | O | |
| | Zoellner et al. (2014) [52] | O | | | | | O | O | Walking coach |

disseminating the research results. Among them, community partners participated as co-authors of research papers in eight studies, in addition to disseminating research results to local residents through flyers or newspapers in only three studies.

**Table 4. Application of CBPR principle.**

| Purpose of study | Author (year) | Planning/execution | | Topic selection/adaptation | | Analysis/ implementation | Dissemination |
|---|---|---|---|---|---|---|---|
| | | Planning | Execution | Topic selection | Adaptation | | |
| Development process | Bess et al. 2019 [38] | O | O | | O | NA | NA |
| | Bradley & Puoane (2007) [35] | O | O | O | | NA | NA |
| | Delisle et al. (2013) [36] | O | O | O | | NA | NA |
| | Perry et al. (2017) [51] | O | O | | O | NA | NA |
| Application/evaluation | Brewer, Balls-Berry et al. 2017 [26] Brewer, Morrison et al. 2019 [27] | O | O | | O | | |
| | Chimberengwa & Naidoo (2019) [46] | | O | | | | |
| | Folta et al. (2019) [49] | | O | | | | |
| | Jayaprakash et al. (2016) [39] | O | O | | | | O |
| | Kim et al. (2004) [40] | | O | | | | |
| | Lynch et al. (2019) [37] | O | O | O | | | O |
| | Mudd-Martin et al. 2013 [41] | O | O | O | | | |
| | Pazoki et al. (2007) [42] | | O | O | | | O* |
| | Pinsker et al. (2017) [43] | O | O | O | | | O |
| | Rorie et al. (2011) [47] | | O | | | | |
| | Schroeder et al. (2017) [48] | O | O | O | | | O |
| | Schulz et al. (2015) [44] | O | O | O | | | O |
| | Skolarus et al. (2018) [45] | O | O | O | | | O, O* |
| | Zoellner et al. (2014) [52] | O | O | O | | | O |
| Development/application/ evaluation | Balcazar, Rosenthal et al. 2009 [23] Balcazar, de Heer et al. 2010 [24] Balcazar, Wise et al. 2012 [25] | O | O | | O | | O* |
| | Brewer, Hayes, Caron et al. 2019 [28] Brewer, Jenkins, Lackore et al. 2018 [29] | O | O | | O | | O |
| | Guzman-Tordecilla (2002) [50] | O | O | | O | | |
| | Ralston, Lemacks et al. 2014 [30] Ralston, Young-Clark et al. 2017 [31] Ralston, Wickrama et al. 2020 [32] | O | O | O | | | |

* Community partners disseminate program results to local residents through flyers or news

*Application of a planning model or health-related theory to program development.* Twelve of the selected studies applied a health planning model or health-related theory during the development process of a CVD management programme. Six studies applied planning models: three studies with intervention mapping, one with the precede–proceed model, and one with the chronic care model.

Seven studies applied a health-related theory, two of which applied two or more theories in combination: four studies applied self-determination theory; three studies, socio-ecological theory; and two studies, transtheoretical models. The theory of planned behaviour, social cognitive theory, health belief model, and theories on self-efficacy, social support, motivation, and social norms were applied in the studies.

*Use of field activists.* In 12 of the selected studies, community activists were recruited to run the programmes. Among them, five studies presented the details of activist education programmes. In a programme for Hispanics, activists (called 'promotores') were mainly recruited

[23–25, 41]. Promotores were selected from among those with participation history in community projects and were asked to obtain health-related qualifications within 30 days of employment. They were trained for a total of 45 hours over two weeks on CVD risk factor reduction methods, capacity reinforcing strategies, community resource identification, advocacy, and data collection methods. Community partners participated in the development of education programmes for promotores. In a study by Kim et al. [40], a Lay Health Advisor (LHA) was recruited to provide a total of 13 education sessions, which took place twice a week (3 hours per session), on physical activity, maintenance of smoking cessation environment, and healthy eating habits. During the last two education sessions, LHAs were given an opportunity to demonstrate methods of practice and personal training. Since LHAs lacked experience in conducting research projects, they were trained in advance on obtaining consent from participants and on collecting data. Delisle et al. [36] selected eight peer fitness trainers and gave them 40 hours of physical activity training (developed by the researchers' team), conducted motivational interviews for behavioural changes, and provided additional training on behavioural change modelling, goal setting, and techniques to enhance capacities.

However, the remaining seven studies only described the role of activists without the details of activist education programmes. Activists were recruited from among residents who spoke the same language as the residents. Their main roles were participant recruitment [26, 31, 47], resident education [47], operational assistance for programmes [26, 31, 42, 44], and a point of contact between participants and researchers during the programme [52]. Some studies that described how they utilised activists, reported paying salaries to some or all activists so that they can fulfil their roles [24, 25, 41, 47].

## The effects of CBPR-based CVD management programmes

Of the 21 selected studies, 17 studies evaluated the effectiveness of the CBPR-based CVD management programme after implementation (Table 5).

The studies were categorised by the study design as follows: 5 studies with one group pre-/ post-test design, 4 RCTs, and 1 non-randomised control group design among quantitative studies, 5 mixed-methods studies, and 2 qualitative studies. The duration of the intervention varied from 4 weeks to 36 weeks, with 24 weeks (6 months) being the most common (5 studies), followed by 8 weeks (3 studies). Regarding programme details, 10 studies explored two or more topics: 15 programmes were on nutrition (or diet) and physical activity, 7 on knowledge related to CVD, 3 on smoking-free habits, 3 on stress management, 2 on weight loss, and 1 on mobile health screening. As for how the programme was run, 9 studies were conducted with counselling (personal contacts, home visits, and calls), 3 studies with multimedia-based education (social networking and text messages), 7 studies with lectures (video and audio tape), 3 studies with cooking demonstrations (models), 8 physical activity programmes (walking, dance, exercise classes, treadmill, and pedometer), 6 small group discussions and seminars, 3 posters, and 1 civic engagement (HEART club).

To measure the effectiveness of each CBPR-based CVD management programme, clinical, behavioural, cognitive, and psychological variables were used; 7 studies measured clinical characteristics (BP, total cholesterol, HDL-cholesterol, LDL-cholesterol, weight, and heart rate); 9 studies measured behavioural characteristics (diet, physical activity, smoke-free, pill pickup rate, treatment compliance, screening rates, follow-up appointments, and participation rates); 5 studies measured changes in CVD-related knowledge as cognitive characteristics; and 1 study measured the quality of life based on psychological characteristics. In addition, 4 studies evaluated research perceptions, benefits, satisfaction, acceptability, and feasibility. In 6 of the

**Table 5. The details of CVD management program based on CBPR.**

| Authors (year) | Study design | Intervention | | |
|---|---|---|---|---|
| | | Session /duration | Name of intervention (detailed contents) | Method and media |
| Brewer, Hayes, Jenkins et al. 2019 [29] | Pre–post | 10 weeks and 28 weeks follow up | Fostering African American Improvement in Total Health (FAITH!) App (nutrition, physical activity) | App-multimedia education modules (videos), self-quizzes, diet/PA self-monitoring, social networking |
| Brewer, Balls-Berry et al. 2017 [26] | Pre–post | 90 minutes/2 weeks, 16 weeks and 3 months follow up | Fostering African American Improvement in Total Health (FAITH!) (cardiovascular health topics, nutrition, physical activity) | Material, interactive lectures, videos, cooking demonstrations, exercise classes |
| Guzman-Tordecilla et al. 2022 [50] | Pre–post | 45 minutes/week, 6 months | Aerobic dance (physical activity) | Supply fruit and water and measure blood pressure. Short messages, expectation campaign. A contest held. Group meeting for promotors. |
| Lynch et al. 2019 [37] | Pre–post | 2 hours, 24 times/9 months | Abundant Living in Vibrant Energy (ALIVE) intervention (nutrition, physical activity) | Bible study, small group sessions, church-wide activities, videos, hand-outs, self-monitor vegetable consumption, bulletin |
| Pinsker et al. 2017 [43] | Pre–post | 12 weeks | Body and Soul program (nutrition) | BP check, cooking demonstrations, peer counselling, video, handbook, posters |
| Schroeder et al. 2017 [48] | Pre–post | 2 hours/week, 2 months (spring, fall) (5 years) | Dance for Health (physical activity) | Line dance |
| Zoellner et al. 2014 [52] | Pre–post | 90 minutes/month, 6 months | HUB City Steps (lifestyle intervention) (nutrition, physical activity) | Peer support, pedometer, group physical activity |
| Balcazar et al. (2010) [24] | RCT | 2 hours/week, 2 months and 2 months follow up | Exp: Su Corazón, Su Vida (nutrition) Con.: Basic educational materials | Exp: telephone calls, small group session Con.: material |
| Folta et al. 2019 [49] | RCT | Exp.: 2 times/week, 24 weeks Con.: 1 time/month, 24 weeks | Exp.: Strong Hearts, Healthy Communities (SHHC) curriculum (diet, nutrition, physical activity) Con.: Didactic healthy lifestyle classes | Exp.: aerobic exercise, progressive strength training, healthy eating practices, discussion + HEART club (civic engagement) Con.: didactic approach |
| Pazoki et al. 2007 [42] | RCT | 1.5 hours/week, 8 weeks | Exp.: Lifestyle modification program (CV diseases, smoking, nutrition, physical activity) Con.: NR | Exp.: audio-taped, material, home-visits, educational packages Con.: NR |
| Ralston, Wickrama et al. 2020 [32] | RCT | 6 months, 12 months follow up and 6 months follow up | Exp.: Health for Hearts United (HHU) intervention (CVD knowledge, diet, stress management, physical activity) Con.: Health ministry development activities (CVD knowledge, nutrition, physical activity, stress reduction) | Exp.: materials, counselling, health promotion activities Con.: materials |
| Schulz et al. 2015 [44] | RCT | 3 times (45–90 minutes)/week, 32 weeks | Exp.: Walk Your Heart to Health (WYHH) intervention (physical activity) Con.: lagged intervention | Exp.: contacts, walking Con.: lagged intervention |
| Rorie et al. 2010 [47] | non RCT | Exp.: summer (2007, 2008) Con.: summer (2007, 2008) | Exp.: Resident health advocate (RHA) intervention (mobile health screening) Con.: standard recruitment | Exp.: personal contacts, flyers Con.: flyers |
| Kim et al. 2004 [40] | Mixed (pre–post, qualitative) | 2 hours/week, 1 month | Health education classes (nutrition, physical activity, smoke-free) | Video, audiotape, picture cards, plastic food models, poster, measuring spoons and cups |
| Mudd-Martin et al. 2013 [41] | Mixed (pre–post, qualitative) | 2 hours/week, 8 weeks | Su Corazon, Su Vida (Your Heart, Your Life) (cardiovascular diseases, nutrition, stress management, physical activity, smoke-free) | Educational models, deep breathing, imagery exercises, discussions |

(*Continued*)

**Table 5.** (Continued)

| Authors (year) | Study design | Intervention | | |
| --- | --- | --- | --- | --- |
| | | Session /duration | Name of intervention (detailed contents) | Method and media |
| Skolarus et al. 2018 [45] | Mixed (RCT, qualitative) | 6 months | Exp.: Reach Out (Mobile Health Technology Intervention) Con.: usual care (hypertension, diet, physical activity) | Exp.: material, BP self-monitoring, text messages Con.: material |
| Chimberengwa & Naidoo 2019 [46] | Qualitative | 1 time/month, 6 months | Chronic care management (health education) | BP check, peer support, counselling, group discussions |
| Jayaprakash et al. 2016 [39] | Qualitative | 60–90 minutes/week, 6 weeks | South Asian Heart Lifestyle Intervention (SAHELI) (heart disease, physical activity, diet, weight, stress management) | Exp.: group classes, activities, counselling, telephone support, pedometers |

selected studies with a randomised controlled group, the experimental group showed significantly positive effects compared to the control group (Table 6).

Qualitative studies cited statements from interviews with participants in the CBPR-based CVD management programme to report their positive evaluation of the programmes [45, 46], improved awareness of CVD, and positive changes in health behaviour [39–41]. Some studies have shown that field activists have the capacity to play an intermediary role between residents and health care professionals [46] and play a key role in ensuring that CBOs recruit and retain participants. In addition, the importance of fostering trust and rapport between programme facilitators and participants as well as the need for family participation were suggested [39, 41]. On the other hand, the lack of sustainability of participation was suggested as a limitation [31].

## Discussion

We reviewed the studies published over the last 10 years on CBPR-based CVD management programmes for adults and aimed to examine the key factors and effects of such programmes and provide fundamental evidence for seeking effective interventions in the future.

In the process of applying CBPR, the following results were derived. First, CBPR partnership is initiated by local residents, local health experts, or researchers. In the studies selected for this review, there were only four studies in which local residents voluntarily proposed research topics. The fact that community partners have become the subject of research initiation means that they were aware of community health problems and active in the process of finding alternative solutions. Whether local residents are the subject of a CBPR initiation is considered an important factor, because there is a higher possibility that it can be implemented as an actual health promotion project [13]. Therefore, in-depth exploration of the context of the community and previous experiences is required for the cases.

In community-based health research, it is recommended or required to establish and operate a community advisory board or steering committee to reflect and communicate the opinions with community members throughout the research process [53]. In previous studies, the role of the participant organisation was limited to a consultation role in which it relayed the opinions and responses of the community. On the other hand, in CBPRs, these act as a representative organisation that provides key information such as the characteristics, strengths, weaknesses, and needs of the local community and play the role of an equal partner in practice during the decision-making process on a problem at hand [54]. For community participation to be considered as community capacity, a global and solid foundation for participation must be formed. It should depart from the typical participation that merely fills up the positions of participants as a formality and instead, be practical such that it involves cooperative

**Table 6. The effectiveness of CVD management program based on CBPR.**

| Authors (year) | Study design | Study participants | Statistical method | Effect (statistical significance) |
|---|---|---|---|---|
| Brewer, Hayes, Jenkins et al. 2019 [29] | Pre–post | Age≥18, N = 50 (mean age: 49.6) | Wilcoxon signed rank test, McNemar test | BP: (+)<br>Behaviours (diet, fruit/vegetable Servings/day, Physical activity): (+)<br>LS7 score: (+) |
| Brewer, Balls-Berry et al. 2017 [26] | Pre–post | Age: 30–75, N = 37 (mean age: 51.7) | Paired t-test, McNemar's test | Increased knowledge: (+)<br>Ideal or intermediate LS7 score: 70% → 82%<br>High LS7 correlated with high psychosocial measure |
| Lynch et al. 2019 [37] | Pre–post | Age≥18, N = 206 (mean age: 57.5) | Paired t-test, Wilcoxon rank-sum test | Vegetable servings consumed/day, total diet quality: (+)<br>Weight and blood pressure: (+) |
| Pinsker et al. 2017 [43] | Pre–post | Participants (n = 310), church coordinators (n = 11) | Chi-squared test, paired t-test | Eating habits (weekly servings of fruit and vegetables): (+)<br>Physical activity: (+)<br>Peer counselling: (+) |
| Schroeder et al. 2017 [48] | Pre–post | Adult: N = 372 (mean age: 52.4)<br>Children: N = 149 (mean age: 12.2) | Mixed effects model, linear or logistic regression | Achieved target heart rate: (+)<br>Activity level during the dance session: (+)<br>High PACES Scales: (+) |
| Zoellner et al. 2014 [52] | Pre–post | Age≥40, N = 269 (mean age: 44.0) | Chi-squared test, two sample t-test, generalized linear mixed model using maximum likelihood estimation | Decrease systolic and diastolic BP: (+)<br>Decreased Sugar intake: (+) |
| Guzman-Tordecilla et al. 2022 [50] | Pre–post | N = 60, (mean age: 28) | Propensity score matching, probit regression model | Decrease systolic and diastolic BP: (+) |
| Balcazar et al. (2009, 2010, 2012) [22–24] | RCT | Exp.: N = 192, (mean age: 53.5)<br>Con.: N = 136 (mean age: 54.0) | Chi-squared test, paired t-test, between-groups analysis of covariance, intention-to-treat analysis | Clinical indicator: all indicator (weight, BP, LDL, HDL, total cholesterol, HbA1c) (-)<br>Nutrition-related behaviour: weight control practice (+), salt intake (+), cholesterol and fat intake (+)<br>Health beliefs: benefits (+), susceptibility (+) |
| Folta et al. 2019 [49] | RCT | Only female<br>Exp.: N = 101 (mean age: 58.9)<br>Con.: N = 93 (mean age: 59.0) | t-test, chi-squared test, multilevel logistic regression: | Diet behaviours (intake of fruit and vegetables): (+)<br>Physical activity (Walking MET- min/week): (+) |
| Pazoki et al. 2007 [42] | RCT | Only female<br>Exp.: N = 179 (mean age: 39.4)<br>Con.: N = 179 (mean age: NR) | Chi-squared test, unpaired t-test, Man-Whitney test | Decrease of systolic BP: (+)<br>Physical activity: (+)<br>Healthy heart knowledge: (+) |
| Ralston, Wickrama et al. 2020 [32] | RCT | Age≥45<br>Exp.: N = 101<br>Con.: N = 110 | Correlation analysis<br>Repeated-measure ANOVA | Fruit and vegetable screener results: (+) |
| Schulz et al. 2015 [44] | RCT | Age≥18, N = 695 (mean age: 46.6)<br>Exp.: NR<br>Con.: NR | Odds ratio with 95% confidence interval, generalized estimating equations | Physical activity: increased steps (+)<br>Indicators of cardiovascular risk: decreased high BP (+), reduction total cholesterol (+), waist circumference (+) |
| Rorie et al. 2010 [47] | non RCT | age≥18<br>Exp.: N = 100 (mean age: 45.1)<br>Con.: N = 47 (mean age: 42.8) | Chi-squared test, t-test, relative risk with 95% confidence interval, | Screening rates: RR 1.55<br>Follow-up appointment: 15% → 55% |
| Kim et al. 2004 [40] | Mixed (pre–post, qualitative) | Age≥18, N = 272 | Quan: Paired t-test<br>Qual: Grouping and identifying theme | Quan: lifestyle behaviours (physical activity, nutrition behaviour, smoke-free behaviour): (+)<br>Qual: success and challenge |
| Mudd-Martin et al. 2013 [41] | Mixed (pre–post, qualitative) | Age≥18, N = 22 (mean age: 44.5) | Quan: Paired t-test<br>Qual: reviewed PI and promotora and identified themes. | Quan: knowledge (CVD and DM): (+)<br>HPLP II scale: (+)<br>Qual: acceptability, responsiveness and difficulty |

(*Continued*)

**Table 6.** (Continued)

| Authors (year) | Study design | Study participants | Statistical method | Effect (statistical significance) |
|---|---|---|---|---|
| Skolarus et al. 2018 [45] | Mixed (RCT, Qualitative) | Age≥18<br>Exp.: N = 48 (mean age: 57.8)<br>Con.: N = 46<br>(mean age: 58.7) | Quan: Univariate statistics, two-sample t-test, Wilcoxon signed rank test<br>Qual: focus group data analysis | Quan: decrease BP (-), texted back 47%, satisfaction 100%<br>Qual: positive responses to tailored text messages, technical issues with responding |
| Chimberengwa & Naidoo 2019 [46] | Qualitative | N = 22 (two focus groups) | Focus groups data analysis | Improvement of knowledge, pickup rate, treatment compliance<br>Importance of training village health workers. |
| Jayaprakash et al. 2016 [39] | Qualitative | Age: 30–60, N = 31 | Thematical analysis | Success: trusted CBO setting, culturally concordant study staff activities, self-monitoring with pedometers helped family involvement and women-only classes were beneficial. Need to reduce participant burden and greater financial resources. |

discussions on planning and assessment of community health promotion projects through active participation with a sense of ownership [55]. However, while most of the selected studies have described the details of partnership establishment, there were only a few studies in which community representatives played a leading role. Therefore, it is necessary to provide opportunities for them to participate in research to exert influence on the community.

The key of CBPR is that members of the local community continuously and actively participate as equal partners with health experts and researchers during the entire process, from research planning to evaluation and reflection of results. Such a process enhances autonomous and sustainable community capacity for health promotion [12]. In this study, for the quality assessment of CBPRs in the selected studies, we partially modified the tool, which was originally developed by Viswanathan et al. [33] and modified by Chen et al. [34], and lays out the core principles of community participation research. In all the studies selected in this study, community partners partially participated during the process of conducting research. However, there were relatively few studies in which they participated in the process of planning and selecting or modifying the research topic. In the case of studies in which they did not participate during the process of planning, but during conducting, the quality of CBPR can be evaluated as low, as community partners mostly participated passively by recruiting participants or assisting in the programme. In particular, none of the studies involved community partners in the analysis or interpretation of the research results. It is necessary to develop strategies, including the development of evaluation tools, to allow community partners to be effectively involved in the process of evaluating research results. A total of 10 studies were conducted in which community partners participated in the dissemination of research results. In three of them, community partners participated in disseminating the research results to local residents through leaflets or newspapers. In the existing conventional health care research system, the low rate of practical implementation has been criticised [56]. However, since CBPR places an emphasis on 'practice' as its core element and has an advantage of offering a possibility of participatory linkages that facilitates dissemination of academic research results to the field, the Prevention Research Center (PRC) of Centers for Disease Control and Prevention (CDC) requires its programme to perform more than one CBPR per core research area [57, 58]. Nevertheless, according to the results of our study, there were very few studies in which community residents participated in the wide dissemination of research results to the community, which is considered one of the merits of CBPR. While an ideal study would satisfy all five criteria of CBPR quality evaluation, none of the selected studies did so. Since inputs from community residents on health projects can make an important contribution to the health promotion

and welfare improvement of the local community and act as an important factor in disease prevention [59], it is necessary to understand the practical reasons why it is difficult for community members to participate in the entire research process from an equal position and apply the true CBPR principles.

Cook [13] proposed to use two complementary methods, a qualitative research method using participatory observation and a quantitative research method using data analysis, for effective CBPR research. Among selected studies, quantitative studies accounted for the largest number at 10; next, there were 6 mixed method studies, and 5 qualitative studies. CBPR is a process-centred research in which qualitative research methods have been proven to be particularly useful, as it emphasises the process of conversation and discussion between community members, researchers, and representatives of public-private organisations, in order to share an understanding of the significance and background of current health problems in the community [12]. In addition, despite having the same research topic, its need and significance can be perceived differently depending on the context and characteristics of each community; a combination of quantitative and qualitative research methods should be used to address these issues. Therefore, the use of mixed-methods research approaches can well reflect the characteristics of CBPR that exhibit a cyclical structure in the process of identifying and solving various problems in communities [13].

An RCT is the gold standard in terms of generating scientific evidence in clinical research; it is conducted with the most rigorous and thorough research methods to explain the relationship between the intervention and an outcome and can provide the highest level of evidence [60]. However, most of the studies selected in our review were non RCTs or pre–post studies; only six studies were RCTs. In the future, to clarify the effectiveness of interventions on CBPR-based CVD management programmes and apply the best evidence to practice, more research should use an RCT design. On the other hand, since the evaluation design with experimental and control groups can contradict the basic premise of CBPR that emphasises the context and participation of the community, a lagged intervention used in Schulz et al. [44] may be considered to solve this issue.

As examined in our review, CBPR-based CVD management programmes showed positive effects on clinical, behavioural, cognitive, relational, and psychological aspects. In particular, Ralston et al. [30–32] showed that there is a longer effect in the experimental group than in the control group through repeated measurements after an intervention with an RCT design. However, while CPBR emphasises strengthening community capacity by allowing community residents to participate in decisions that affect their own health and exert influence on the quality of life of the entire community by establishing cooperative partnerships, securing sufficient resources, and promoting active participation and leadership of community members [61], none of the selected studies in our review measured the effects on community capacity. Specifically, measuring the effectiveness should not end with a one-time measurement immediately after the intervention; instead, it should evaluate the sustainability of an intervention and reflect the CBPR evaluation criteria suggested by Plumb et al. [62], including cooperation of community, group dynamics, and evaluation of community participation.

This scoping review for the key elements and effects of CBPR-based CVD management programmes over the last 12 years examined the recent trends in such programmes. Our results examined the sustainability and applicability of the CBPR-based CVD management programme by allowing community partnerships to understand the health needs of community residents, develop and implement programmes that are acceptable to community residents, and confirm for their effectiveness, instead of using the unilaterally provided standardised programmes.

This review has the following limitations: first, as our review included only studies published in English, most of the selected studies were conducted in North America, which did not reflect the cultural diversity of non-English speaking countries; and, second, with a focus on the development, implementation, and evaluation of CVD management programmes, problems such as obstacles and challenges arising in some processes of CBPR could not be emphasised.

## Conclusions

We examined the key elements and effects of CBPR-based CVD management programmes using the scoping review method. Among the key elements of such programmes, there were many cases in which community organisations (CBOs, FBOs) played the leading role in establishing partnerships and cases in which a decision-making committee was formed. Regarding the application of the CBPR principles, community partners rarely participated during the process of analysis and interpretation of research results and dissemination of research results to residents. Of the 17 studies in which the program effects were evaluated, all of the six RCT studies showed significantly positive effects in the experimental groups as compared to the control groups.

It is necessary to develop strategies to improve the criteria based on which CBPR principles cannot be implemented during the development and application of a CBPR-based CVD management programme. In addition, more attempts to verify the effectiveness of high-quality research methodology should be made when evaluating the effectiveness of programmes.

## Supporting information

**S1 Appendix. Search strategies by electronic databases.**
(DOCX)

**S1 Checklist. Preferred reporting items for systematic reviews and meta-analyses extension for scoping reviews (PRISMA-ScR) checklist.**
(DOCX)

## Acknowledgments

We would like to thank PhD student Eun-shim Kim, who participated in the protocol work of this study. In addition, We would like to appreciate the advice from Professor Baek-geun Jung, Yoon-sik Kang, and Young-mi Ha.

## Author Contributions

**Conceptualization:** Juhyeon Yang, Bohyun Park.

**Data curation:** Juhyeon Yang, Bohyun Park.

**Funding acquisition:** Bohyun Park.

**Methodology:** Bohyun Park.

**Writing – original draft:** Juhyeon Yang, Bohyun Park.

**Writing – review & editing:** Bohyun Park.

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
