## [Decision Letter · Decision Letter 0]

24 Mar 2022

PONE-D-21-11065A scoping review on the key elements and effects of cardiovascular disease management programs based on community-based participatory researchPLOS ONE

Dear Dr. Park,

Thank you for submitting your manuscript to PLOS ONE. After careful consideration, we feel that it has merit but does not fully meet PLOS ONE’s publication criteria as it currently stands. Therefore, we invite you to submit a revised version of the manuscript that addresses the points raised during the review process.

Literature search results are "results" not methods. The flow diagram and literature search results/numbers should be in the first paragraph of results, not methods. Further, please, address the reviewers' comments below.

We look forward to receiving your revised manuscript.

Kind regards,

Ahmed Negida, MD

Academic Editor

PLOS ONE

Journal Requirements:

2. During the internal evaluation of your manuscript we have noted that the literature search was completed in April 2020. Please consider whether an update to the literature search should be conducted since scoping reviews should provide an up-to-date overview of the research topic covered.

4. Your abstract cannot contain citations. Please only include citations in the body text of the manuscript, and ensure that they remain in ascending numerical order on first mention.

Reviewers' comments:

Reviewer's Responses to Questions

**Comments to the Author**

1. Is the manuscript technically sound, and do the data support the conclusions?

Reviewer #1: Yes

Reviewer #2: Yes

2. Has the statistical analysis been performed appropriately and rigorously? 

Reviewer #1: N/A

Reviewer #2: N/A

3. Have the authors made all data underlying the findings in their manuscript fully available?

Reviewer #1: Yes

Reviewer #2: Yes

4. Is the manuscript presented in an intelligible fashion and written in standard English?

Reviewer #1: Yes

Reviewer #2: Yes

5. Review Comments to the Author

Reviewer #1: Thanks for your efforts regarding this well written review summarizing current CBPR practices, the research question is an important one in the current Cardiology community practice which was properly addressed by the authors, also results led to proper conclusions

I had some comments regarding

Abstract :

Any reason for choosing the age above 18 to be enrolled in the review?

It was written in the abstract in all 6 RCTs without any brief explanation of them in previous paragraph.

Materials and methods:

Describe review method developed by Arcey and O'Malley.

Describe PRISMA extension for scoping review check list.

Please explain search terms Cardiovasc* , vascul* disease, PAR ? , participat ?

In line 270, 271 describe each model

Details of statistics of your chosen studies need to be properly high lighted.

Rewrite both tables 2 and 3 (establishing partner ship, CBPR principle) in a simple fashion.

Reviewer #2: The authors presented a scoping review about e key elements and effects of cardiovascular disease

management programs based on community-based participatory research. The idea is interesting, the article is well written. I thank the authors for their effort. However, minor concerns exist.

1-In table 1, there is a study where adults & children were included and another one that did not report the included age. This does not meet the criteria of your selection (>18 y).

2-The introduction and the discussion need to be shortened.

3-The answers for the research questions should be clarified during the discussion.

6. PLOS authors have the option to publish the peer review history of their article (what does this mean?). If published, this will include your full peer review and any attached files.

Reviewer #1: No

Reviewer #2: **Yes: **Mohammad Eltahlawi

---

## [Author Response · Author response to Decision Letter 0]

11 Jul 2022

Authors’ Response for the Reviewers’ Report:

# Editor

Thank you for your good comments on this study. You can check the revised contents through the track change.

1. Literature search results are "results" not methods. The flow diagram and literature search results/numbers should be in the first paragraph of results, not methods.

Generally, in scoping reviews and systematic reviews, the study selection process is described in the research method.

After reviewing PLOS ONE’s style requirements, we revised the manuscript and references.

3. During the internal evaluation of your manuscript we have noted that the literature search was completed in April 2020. Please consider whether an update to the literature search should be conducted since scoping reviews should provide an up-to-date overview of the research topic covered.

After searching for additional published studies from April 2020 to April 2022, the results were added and described.

4. In your Data Availability statement, you have not specified where the minimal data set underlying the results described in your manuscript can be found. PLOS defines a study's minimal data set as the underlying data used to reach the conclusions drawn in the manuscript and any additional data required to replicate the reported study findings in their entirety.

In this study, the data extraction process was presented in detail in the research method. Tables 2–5 present the result of the study, that is, the final data. The search strategies were attached as supplementary data.

5. Your abstract cannot contain citations. Please only include citations in the body text of the manuscript, and ensure that they remain in ascending numerical order on first mention.

The abstract does not contain any citations any longer.

6. Please include captions for your Supporting Information files at the end of your manuscript, and update any in-text citations to match accordingly. 

We attached the search strategies for each database and PRISM-ScR at the end of our manuscript.

7. Please review your reference list to ensure that it is complete and correct. 

We reviewed and revised the reference list.

Reviewer #1:

Thank you for your good comments on this study. You can check the revised contents through the track change.

Thanks for your efforts regarding this well written review summarizing current CBPR practices, the research question is an important one in the current Cardiology community practice which was properly addressed by the authors, also results led to proper conclusions

Thank you for your encouraging comments. We have made revisions based on your feedback, which, we believe, has improved the quality of our paper.

1. Abstract: Any reason for choosing the age above 18 to be enrolled in the review?

The target population of this study is adults. Many studies define people over the age of 18 as adults, and thus we applied that criterion in this study.

2. Abstract: It was written in the abstract in all 6 RCTs without any brief explanation of them in previous paragraph.

We added a brief explanation of this point, as follows.

Page 2 Line 35 to 37

In addition, among the 21 studies selected were 6 RCTs, all of which showed a significant positive effect in experimental groups compared to control groups.

3. Materials and methods: Describe review method developed by Arcsey and O'Malley.

The review method developed by Arcsey and O’Malley is described in lines 96–100, on page 6.

4. Describe PRISMA extension for scoping review check list.

PRISMA-SCR is an abbreviation of ‘Preferred Reporting Items for Systematic Reviews and Meta-Analyses Extension for Scoping Reviews’, and the full term is presented inside parentheses.

Page 5 Lines 98 to 99: In the process of conducting this scoping review and reporting the results, we abided by the PRISMA-ScR (Preferred Reporting Items for Systematic Reviews and Meta-Analyses Extension for Scoping Reviews) checklist [22].

5. Please explain search terms Cardiovasc* , vascul* disease, PAR ? , participat ?

* is a symbol that has the function to search for all words that match the spelling located before * when searching in electronic databases such as PubMed and CINAHL (Searching using 'Cardiovasc*' searches for 'Cardiovascular', 'Cardiovasculitis', etc). PAR, which is abbreviation for Participatory Action Research, was selected in the search because it is a research form similar to CBPR. ‘participat*’ was revised to ‘participant*’ because it was a typo.

6. In line 270, 271 describe each model

‘Intervention mapping’ and ‘precede–proceed model’ are the names of the health planning models. Since these models were used in the selected studies, we simply listed the names of models in the manuscripts. We deleted 'CATCH-PATH model' as it was confirmed that it was not a planning model.

7. Details of statistics of your chosen studies need to be properly highlighted.

The statistical methods used in the selected study were added to Table 5.

8. Rewrite both tables 2 and 3 (establishing partnership, CBPR principle) in a simple fashion.

The text of Tables 2 and 3 has been restated more concisely. Lines 227–247.

Reviewer #2: 

Thank you for your good comments on this study. You can check the revised contents through the track change.

The authors presented a scoping review about key elements and effects of cardiovascular disease management programs based on community-based participatory research. The idea is interesting, the article is well written. I thank the authors for their effort

Thank you for your encouraging comments. We have made revisions based on your feedback, which, we believe, has improved the quality of our paper.

1. In table 1, there is a study where adults & children were included and another one that did not report the included age. This does not meet the criteria of your selection (>18 y).

Although Schroeder et al. (2017) studied both adults and children, we included results from the study with adults because we were able to confirm them. We deleted “children” from the table because it could make confusion for readers.

2. The introduction and the discussion need to be shortened.

We reviewed the introduction and discussion parts and described them more concisely. We deleted the description of why we used the scoping review method at the beginning of the discussion.

3. The answers for the research questions should be clarified during the discussion.

The discussion was revised to focus on the answers to the research questions.

---

## [Decision Letter · Decision Letter 1]

3 Oct 2022

PONE-D-21-11065R1A scoping review on the key elements and effects of cardiovascular disease management programs based on community-based participatory researchPLOS ONE

Dear Dr. Park,

Thank you for submitting your manuscript to PLOS ONE. After careful consideration, we feel that it has merit but does not fully meet PLOS ONE’s publication criteria as it currently stands. Therefore, we invite you to submit a revised version of the manuscript that addresses the points raised during the review process.

We look forward to receiving your revised manuscript.

Kind regards,

Julie Gleason-Comstock

Academic Editor

PLOS ONE

Journal Requirements:

Additional Editor Comments:

Abstract/Line 111: Spell out Randomized Controlled Trial before acronym of "RCT"

-Lines 201-203: The new Table 2 including racial statistics includes six studies which authors state did not mention race, and they use the term "Nor." Further author investigation of those six studies as to participant race and explanation of "Nor" is requested.

-Line 420: With author manuscript revisions, the search timeline was expanded from ten to twelve years. Please update.

Reviewers' comments:

Reviewer's Responses to Questions

**Comments to the Author**

1. If the authors have adequately addressed your comments raised in a previous round of review and you feel that this manuscript is now acceptable for publication, you may indicate that here to bypass the “Comments to the Author” section, enter your conflict of interest statement in the “Confidential to Editor” section, and submit your "Accept" recommendation.

Reviewer #1: All comments have been addressed

Reviewer #2: All comments have been addressed

2. Is the manuscript technically sound, and do the data support the conclusions?

Reviewer #1: Yes

Reviewer #2: Yes

3. Has the statistical analysis been performed appropriately and rigorously? 

Reviewer #1: Yes

Reviewer #2: Yes

4. Have the authors made all data underlying the findings in their manuscript fully available?

Reviewer #1: Yes

Reviewer #2: Yes

5. Is the manuscript presented in an intelligible fashion and written in standard English?

Reviewer #1: Yes

Reviewer #2: Yes

6. Review Comments to the Author

Reviewer #1: Thanks for addressing the requested changes and questions, kindly you need to edit participat to participant as it is a typo

also the search timing need to emphasize it was between March and April and data from studies till 2022 were added as its review of literature so we ensure that its updated.

Reviewer #2: The authors addressed the concern well. I Thank you for your satisfactory reply. The discussion now sounds good

7. PLOS authors have the option to publish the peer review history of their article (what does this mean?). If published, this will include your full peer review and any attached files.

Reviewer #1: No

Reviewer #2: **Yes: **Mohammad Eltahlawi

---

## [Author Response · Author response to Decision Letter 1]

7 Nov 2022

Thank you for your good comments on this study. We marked the modified part in red.

We checked the reference list and corrected. No. 27 was missed and added. Number errors (27 -> 28, 28 -> 29) were corrected and No 29 was deleted

2. Abstract/Line 111: Spell out Randomized Controlled Trial before acronym of "RCT"

The ‘RCT’ was deleted from the abstract and presented as 'Randomized Controlled Trial'.

3. Lines 201-203: The new Table 2 including racial statistics includes six studies which authors state did not mention race, and they use the term "Nor." Further author investigation of those six studies as to participant race and explanation of "Nor" is requested.

-> ‘Nor reported’ was corrected to ‘Not reported.’ Six studies that did not mention race were reviewed again and the selected subject characteristics were added.

4. Line 420: With author manuscript revisions, the search timeline was expanded from ten to twelve years. Please update.

I updated search timeline from 10 years to 12 years.

---

## [Editor Report · Decision Letter 2]

12 Dec 2022

A scoping review on the key elements and effects of cardiovascular disease management programs based on community-based participatory research

PONE-D-21-11065R2

Dear Dr. Park,

We’re pleased to inform you that your manuscript has been judged scientifically suitable for publication and will be formally accepted for publication once it meets all outstanding technical requirements.

Kind regards,

Julie Gleason-Comstock

Academic Editor

PLOS ONE
---

## [Editor Report · Acceptance letter]

19 Dec 2022

PONE-D-21-11065R2 

A scoping review key elements and effects of cardiovascular disease management programs based on community-based participatory research 

Dear Dr. Park:

I'm pleased to inform you that your manuscript has been deemed suitable for publication in PLOS ONE. Congratulations! Your manuscript is now with our production department. 

Kind regards, 

on behalf of

Professor Julie Gleason-Comstock 

Academic Editor

PLOS ONE